# Functional Characterization of Transgenic Mice Overexpressing Human 15-Lipoxygenase-1 (ALOX15) under the Control of the aP2 Promoter

**DOI:** 10.3390/ijms24054815

**Published:** 2023-03-02

**Authors:** Dagmar Heydeck, Christoph Ufer, Kumar R. Kakularam, Michael Rothe, Thomas Liehr, Philippe Poulain, Hartmut Kuhn

**Affiliations:** 1Department of Biochemistry, Charité—Universitätsmedizin Berlin, Corporate Member of Freie Universität Berlin, Humboldt Universität zu Berlin, and Berlin Institute of Health, Charitéplatz 1, D-10117 Berlin, Germany; 2Lipidomix GmbH, Robert-Roessle-Str. 10, D-13125 Berlin, Germany; 3Institute of Human Genetics, Jena University Hospital, Friedrich Schiller University, Am Klinikum 1, D-07747 Jena, Germany; 4Genfit, Parc Eurasanté 885, Avenue Eugène Avinée, 59120 Loos, France

**Keywords:** eicosanoids, polyenoic fatty acids, inflammation, oxidative stress, inflammation, atherosclerosis

## Abstract

Arachidonic acid lipoxygenases (ALOX) have been implicated in the pathogenesis of inflammatory, hyperproliferative, neurodegenerative, and metabolic diseases, but the physiological function of ALOX15 still remains a matter of discussion. To contribute to this discussion, we created transgenic mice (aP2-ALOX15 mice) expressing human ALOX15 under the control of the aP2 (adipocyte fatty acid binding protein 2) promoter, which directs expression of the transgene to mesenchymal cells. Fluorescence in situ hybridization and whole-genome sequencing indicated transgene insertion into the E1-2 region of chromosome 2. The transgene was highly expressed in adipocytes, bone marrow cells, and peritoneal macrophages, and ex vivo activity assays proved the catalytic activity of the transgenic enzyme. LC-MS/MS-based plasma oxylipidome analyses of the aP2-ALOX15 mice suggested in vivo activity of the transgenic enzyme. The aP2-ALOX15 mice were viable, could reproduce normally, and did not show major phenotypic alterations when compared with wildtype control animals. However, they exhibited gender-specific differences with wildtype controls when their body-weight kinetics were evaluated during adolescence and early adulthood. The aP2-ALOX15 mice characterized here can now be used for gain-of-function studies evaluating the biological role of ALOX15 in adipose tissue and hematopoietic cells.

## 1. Introduction

Lipoxygenases are fatty acid dioxygenases that oxygenate arachidonic acid and related polyenoic fatty acids to the corresponding hydroperoxy derivatives [1,2,3,4,5]. They have been implicated in the differentiation of mesenchymal [6,7,8,9] and ectodermic cells [10,11,12] but may also play a role in the pathogenesis of inflammatory [13,14], hyperproliferative [15,16,17,18], neurodegenerative [19,20,21], and metabolic [22,23,24,25] diseases. The human genome involves six functional ALOX genes (*ALOX15* [26], *ALOX15B* [27], *ALOX12* [28,29], *ALOX12B* [30], *ALOX5* [31,32], *ALOXE3* [12,33]), and each of the ALOX-isoforms exhibit distinct biological functions. In the mouse genome, a single-copy ortholog exists for each human *ALOX* gene, but in addition, an *Aloxe12* gene exists that encodes for a functional epidermal Aloxe12 [34]. This enzyme shares a high degree of amino acid identity with Alox15, but in humans, the corresponding ortholog is a corrupted pseudogene [34]. Except for ALOXE12, knockout mice are available for each ALOX-isoform ([11,35,36,37,38,39,40]), but despite the availability of these tools, the biological relevance of the different ALOX-isoforms is still a matter of discussion.

Together with ALOX5, ALOX15 is the most comprehensively characterized ALOX-isoform [26,41,42,43]. It was discovered in 1975 as a protein in immature red blood cells of rabbits that was capable of oxygenating mitochondrial membranes [44]. Because of this property, the enzyme has been implicated in the maturational breakdown of mitochondria during late erythropoiesis [45,46]. To explore the biological roles of Alox15 in vivo, ALOX15^−/−^ mice have been generated [35], and these animals (loss-of-function strategy) have been tested in a large number of mouse models of human diseases [47,48,49,50,51,52,53,54]. In addition, a number of transgenic mouse lines have been created (gain-of-function strategy), in which mouse or human ALOX15 was overexpressed under the control of different regulatory elements, that exhibit interesting phenotypes. Moderate overexpression of the endogenous mouse Alox15 induced spontaneous formation of aortic fatty streaks, which was related to upregulated expression of endothelial cell-adhesion molecules [55]. Endothelium-specific overexpression of human ALOX15 [56] accelerated aortic lipid deposition in LDL-receptor deficient mice [57], but it also inhibited tumor growth and metastasis in two different mouse models of human cancer [58]. More recently, these authors showed that this protective effect may be related to the promotion of apoptosis and necrosis in primary and metastatic tumor cells [59]. In rabbits, overexpression of human ALOX15 under the control of the lysozyme promoter induced macrophage-specific expression of the transgene [60] and protected the animals from aortic lipid deposition when fed a lipid-rich Western-type diet [61]. Transgenic mice overexpressing human ALOX15 under the control of the scavenger receptor A promoter [62] were also protected from aortic lipid deposition, but here, the protective effect was related to the augmented biosynthesis of anti-inflammatory and pro-resolving lipid mediators such as lipoxin A4, resolvin D4, and protectin D1 [63]. Genetically modified mice overexpressing a transgenic version of the endogenous Alox15 under the control of the alpha-cardiac myosin heavy chain promoter developed heart failure and diabetic cardiomyopathy [64,65]. When human ALOX15 was overexpressed in the intestinal epithelium under the control of the villin promoter [66], the resulting transgenic mice were protected from the development of azoxymethane-induced colonic tumors, and expression of the ALOX15 transgene was always impaired in tumor cells when compared with non-tumor controls [67]. Additional mechanistic studies have suggested that expression of the transgene inhibited the expression of tumor necrosis factor alpha and its target, the inducible nitric-oxide synthase. Moreover, activation of nuclear factor kappa B was prevented [67]. More recently, a transgenic mouse line was created in which the expression of the human ALOX15 was controlled by the Cre-lox promoter [68]. The employed strategy ensured ubiquitous overexpression of the transgene, but when these mice and corresponding wildtype controls were used in a diabetic peripheral neuropathy model, the authors did not observe significant differences when compared with wildtype controls [68].

In most of these transgenic animals, tissue-specific expression of the transgene was explored. However, incorporation of the transgene into the host genome was controlled in neither of them, and thus, it is unclear how many copies of the transgene had been incorporated into the host genome and at which positions. Moreover, in many of these studies, activity assays were performed, and thus, it is unclear whether the transgenic enzyme was catalytically active. This is a serious limitation for some of these studies, since expression of a functional ALOX15 is strongly regulated on the translational level [69,70,71], and thus, detection of the transgenic mRNA is not sufficient to conclude the catalytic activity of the enzyme. In fact, in human umbilical vein endothelial cells, high levels of ALOX15 mRNA were detected, but the catalytically active enzyme was missing [72]. To contribute to the discussion on the biological role of Alox15, we here characterize transgenic mice expressing the human ALOX15 under the control of the aP2 (adipocyte fatty acid binding protein-2) promoter (aP2-Alox15 mice). In these mice, transgene expression is directed to mesenchymal cells, particularly to adipocytes, macrophages, and other cells of the myeloic linage. In the present paper we report the breeding of homozygous aP2-ALOX15 mice and found that the transgene was incorporated as single copy gene into the E1-2 region of chromosome 2. Ex vivo activity assays indicated expression of the functional transgene in adipocytes, spleen, bone marrow cells, and peritoneal macrophages. The in vivo activity of transgenic human ALOX15 was indicated by analysis of the plasma oxylipidomes. Because of the high expression levels of the transgene in adipose tissue and in the hematopoietic system, these mice can be used in the future to study the role of ALOX15 in adipocyte differentiation and hematopoiesis.

## 2. Results

### 2.1. Creation of Transgenic aP2-ALOX15 Mice

Different ALOX-isoforms (ALOX5 [73], ALOX12 [74], ALOX12B [75], ALOXE3 [76] and ALOX15 [77]) have been implicated in adipocyte differentiation, in the energy metabolism of fat cells, and in adipose tissue remodeling. Moreover, supplementation studies with ALOX products suggested that several oxylipins activate adipogenesis of 3T3 cells in vitro, and these data support a possible role of ALOX15 in adipogenesis [9]. However, 15-HETE formation in mice is limited, since none of the seven functional Alox isoforms produce 15-HETE as a major arachidonic acid oxygenation product. Human ALOX15 effectively oxygenates arachidonic acid to 15-HETE [78,79], and thus, 15-HETE formation can be used as metabolic footprint for expression of the transgene.

To test the impact of endogenous 15-HETE formation in mouse adipose tissue in vivo, a transgenic mouse line was created that overexpresses human ALOX15 under the control of the aP2 (adipocyte fatty acid binding protein 2) promoter. For this purpose, the cDNA of human ALOX15 was ligated behind the aP2 promoter (Figure 1), and this construct was microinjected into fertilized eggs. Cells were reimplanted into pseudo-pregnant mice, and individuals carrying the transgene in the germ line were crossed with wildtype C57BL/6 mice. Heterozygous allele carriers were intercrossed, and homozygous aP2-ALOX15 mice were selected. These animals were used to establish a colony of homozygous aP2-ALOX15 mice, and individuals of this colony were used for our characterization studies.

### 2.2. Genomic Insertion of the Transgene

Since our transgenic strategy involved coincidental incorporation of the transgene into the host genome, we next explored how many copies of the transgene were incorporated into the genome, and we also determined the site(s) of transgene insertion.

For this purpose, we first performed fluorescence in situ hybridization (FISH) using the human ALOX15 cDNA as a probe. Figure 2A shows a representative FISH staining of a heterozygous founder. It can be seen that the transgene was inserted as a single-copy gene into the subband E1-2 on chromosome 2. No specific fluorescence signal was detected on any other chromosome. To describe the site of transgene incorporation more precisely and to exclude transgene insertion having disrupted a functional gene in this region, we carried out whole-genome sequencing. The obtained sequence data confirmed single incorporation of the transgene and also suggested that no functional gene was disrupted during transgene insertion.

### 2.3. Tissue-Specific Expression of Transgenic Human ALOX15

To explore the tissue-specific expression of both endogenous mouse Alox15 and the human ALOX15 transgene, we carried out qRT-PCR using isofom-specific primers. Human and mouse ALOX15 cDNA share a high degree of nucleotide sequence identity (85%), and thus, designing ortholog specific pPCR primers was somewhat difficult. However, we selected cDNA regions with relatively low degrees of nucleotide sequence conservation, and by using these ortholog specific primer pairs, we found (Figure 3A) that the endogenous arachidonic acid 12-lipoxygenating mouse Alox15 was expressed at relatively low levels in most cells and tissues.

The highest expression levels were observed in the lung, but even in this organ, only about 20 copies of Alox15 mRNA were present per 1000 copies of Gapdh mRNA. Lower expression levels were observed in bone marrow cells and in peri-epididymal adipose tissue. In spleen, heart, liver, kidney, and brain, we did not see specific amplification signals. When similar analyses were carried out with the RNA extracts prepared from corresponding tissues of the transgenic aP2-ALOX15 mice, lower steady-state ALOX15 mRNA concentrations were detected in the lungs, but elevated levels were detected in the perirenal adipose tissue (Figure 3A). However, despite these differences, the expression levels of the endogenous mouse Alox15 mRNA were also low (1–5 Alox mRNA copies per 1000 copies of Gapdh) in all tested tissues of the transgenic aP2-ALOX15 mice.

When we repeated these analyses with the human ALOX15-specific amplification primers (Figure 3A), we did not see specific amplification signals in the different tissues of wildtype mice. These results were expected, since wildtype mice do not express human ALOX15, and our primers do not pick up mouse Alox15 mRNA. In contrast, we detected the abundant expression of human ALOX15 in the three types of adipose tissue (peri-epididymal, subcutaneous, perirenal). We also detected high-level expression of the transgene in bone marrow cells, spleen, lungs, and testis. The most interesting outcome of our qRT-PCR data was that the steady state concentrations of the transgenic mRNA, which varied between 800 and 3,500 ALOX15 mRNA copies per 10^3^ copies of Gapdh, were much higher than the mRNA copy numbers of the endogenous Alox15 in wildtype tissues. Taken together, these data indicate overexpression of the transgenic enzyme in adipose tissue, but also in hematopoietic cells (bone marrow), spleen, lung, and testis.

### 2.4. Transgenic Human ALOX15 Is Catalytically Active in Peritoneal Lavage and Bone Marrow Cells

Expression of ALOX15 orthologs is strongly regulated at transcriptional [80,81] and translational levels [69,70]. In fact, in young rabbit reticulocytes, large amounts of ALOX15 mRNA are present, but no functional protein can be detected [45]. Thus, in principle, there is the possibility that ALOX15 mRNA is present but no functional enzyme is expressed [72]. To explore whether a functional transgenic enzyme is expressed in different tissues of the transgenic aP2-ALOX15 mice, we carried out ex vivo activity assays using intact cell suspensions or tissue homogenate supernatants as enzyme source. In peritoneal lavage cells, mouse ALOX15 is highly expressed, and thus, these cells are well-suited to ex vivo activity assays.

When we incubated these cells from wildtype mice with arachidonic acid (Figure 4A), large amounts of 12-HETE were detected. In addition, smaller amounts of 15-HETE were also identified as a minor side product. Interestingly, under these experimental conditions, we did not find any 5-HETE formation, although significant amounts of ALOX5 mRNA were detected in these cells by qRT-PCR.

**Figure 4 ijms-24-04815-f004:**
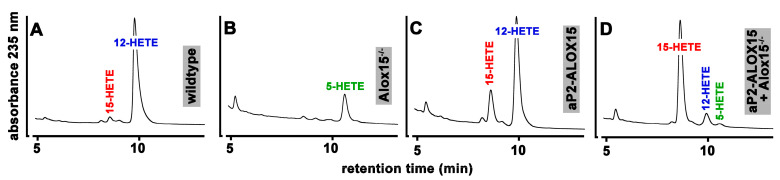
Ex vivo activity assays of peritoneal lavage cells prepared from mice of different genotype. 10^7^ peritoneal macrophages prepared from mice of different genotype were incubated with arachidonic acid as described in Section 4, and the oxygenation products were analyzed by RP-HPLC. Representative chromatograms are shown; the statistical evaluation of the data is given in Table 1.

**Table 1 ijms-24-04815-t001:** Ex vivo activity assays of peritoneal lavage cells prepared from mice of different genotype. 10^7^ peritoneal lavage cells prepared from mice of different genotypes (n = 3 of each genotype) were incubated with exogenously added arachidonic acid as described in Section 4. The arachidonic-acid-derived oxygenation products were prepared and further analyzed by RP-HPLC. The statistical evaluation of the experimental raw data was performed using the Student’s t-test function of the Microsoft Excel software package. ^#^ wildtype mice vs. aP2-ALOX15 animals, *p* < 0.01, ^§^ aP2-ALOX15 mice vs. aP2-ALOX15 + ALOX15^−/−^ animals *p* < 0.01.

Genotype	Relative Share of Different HETE Isomers (%)		
	15-HETE	12-HETE	5-HETE
wildtye	6.7 ± 1.4 ^#^	93.3 ± 1.4 ^#^	<1
ALOX15^−/−^	<1	<1	99.0
aP2-ALOX15	22.1 ± 4.8 ^#,§^	77.9 ± 4.8 ^#,§^	<1
aP2-ALOX15 + ALOX15^−/−^	88.3 ± 1.0 ^§^	10.9 ± 0.8 ^§^	<1

When these experiments were repeated with peritoneal macrophages prepared from ALOX15^−/−^ mice, 12- and 15-HETE were no longer detectable. Instead, 5-HETE was identified as a major arachidonic acid oxygenation product (Figure 4B). Since ALOX5 is expressed in peritoneal macrophages, the formation of 5-HETE is plausible when the dominant ALOX15 pathway is genetically silenced. When we incubated peritoneal lavage cells from our transgenic aP2-ALOX15 mice with arachidonic acid, 12-HETE remained the major oxygenation product (Figure 4C). However, the relative share of 15-HETE was significantly increased when compared with wildtype cells (Table 1). The most plausible explanation for these data is that both endogenous ALOX15 (forming 12-HETE) and the transgenic ALOX15 (forming 15-HETE) were catalytically active. To provide more compelling evidence for this conclusion, we crossed ALOX15^−/−^ mice with our transgenic aP2-ALOX15 animals, prepared peritoneal lavage cells from the animals, and carried out ex vivo activity assays. In these cells, endogenous Alox15 (forming 12-HETE) was absent, and thus, 15-HETE originating from the transgenic ALOX15 pathway was expected to be dominant. In fact, when we carried out such ex vivo activity assays, 15-HETE was the major arachidonic acid oxygenation product, and small amounts of 12-HETE were also detected (Figure 4D). Since human ALOX15 exhibits dual-reaction specificity [79], the formation of small amounts of 12-HETE by the transgenic enzyme is plausible. Similar ex vivo activity assays were carried out with three individuals of each genotype; the statistical evaluation of the experimental raw data is given in Table 1. In summary, these data confirm our hypothesis that the transgenic human ALOX15 is expressed as a catalytically active enzyme in peritoneal macrophages in addition to the endogenous mouse Alox15. Assuming a similar specific activity of mouse and human ALOX15, more endogenous Alox15 should be present in these cells when compared with the transgenic human ALOX15.

Bone marrow cells are another rich source of endogenous Alox15. However, in these cell types, the arachidonic acid metabolism is somewhat more complex, since the arachidonic acid 12-lipoxygenating ALOX15, the arachidonic acid 12-lipoxygenating ALOX12, and the arachidonic acid 5-lipoxygenating ALOX5 are simultaneously expressed. When wildtype bone marrow cells are incubated with arachidonic acid, 12-HETE was identified as a major arachidonic acid oxygenation product (Table 2). 15-HETE and 5-HETE only contributed minor shares. When bone marrow cells of ALOX15^−/−^ mice were used for the ex vivo activity assays, we did not see any 15-HETE formation (Table 2). These data suggest that the minor share of 15-HETE formation by wildtype bone marrow cells (4.9 ± 1.6%) may be related to the catalytic activity of the endogenous mouse Alox15. Although mouse Alox15 is dominantly arachidonic acid 12-lipoxygenating, 15-HETE is a minor side product [82].

The strong but incomplete reduction of 12-HETE formation by ALOX15^−/−^ bone marrow cells suggests that the dominant 12-HETE formation by wildtype bone marrow cells (93.8 ± 3.6%) may be related to the mixed catalytic activity of the endogenous ALOX15 and ALOX12 isoforms. Interestingly, ALOX15^−/−^ bone marrow cells produce large amounts of 5-HETE, and these data (Table 2) are consistent with the results of the ex vivo activity assays of peritoneal lavage cells (Figure 4B, Table 1). Expression of transgenic human ALOX15 completely altered the pattern of arachidonic acid oxygenation. In this case, 15-HETE was the major (55.5 ± 1.6%) arachidonic acid oxygenation product; these activity data suggest catalytic activity from the transgenic human ALOX15. When we crossed aP2-ALOX15 transgenic mice with ALOX15^−/−^ animals, the relative share of 12-HETE was further reduced (from 44.4 ± 1.6% in aP2-ALOX15 mice to 10.9 ± 0.5% in aP2-ALOX15 + ALOX15^−/−^); these data suggest that more than 70% of the 12-HETE formed by aP2-ALOX15 bone marrow cells originated from the endogenous ALOX15 pathway. In these cells, ALOX12 may only contribute 30% to 12-HETE formation. As expected, formation of 15-HETE was strongly elevated by bone marrow cells of aP2-ALOX15 + ALOX15^−/−^ mice. Taken together, our ex vivo activity experiments confirm that in peritoneal lavage cells, as well as in bone marrow cells, transgenic human ALOX15 is expressed and the transgenic enzyme is catalytically active.

### 2.5. Transgenic Human ALOX15 Is also Catalytically Active in Solid Tissue

As indicated in Figure 3, the transgenic mRNA is expressed in different solid tissues such as adipose tissue, spleen, lungs, and testis. To explore whether the transgenic enzyme is also expressed in these tissues and whether the protein is catalytically active, we carried out similar ex vivo activity assays. For this purpose, we prepared tissue homogenates, and used the 20,000 g supernatants as enzyme source. After a 15 min incubation period of the homogenate supernatants with arachidonic acid, we quantified by RP-HPLC the formation of 12-HETE and 15-HETE as the major readout parameter.

When homogenate supernatants of wildtype control mice were used for the ex vivo activity assays, only small amounts of 12-HETE and 15-HETE were formed. Only in lungs and spleen, we observed significant formation of 12-HETE, which exceeded the formation of 15-HETE. These data suggest that the endogenous arachidonic acid 12-lipoxygenating mouse Alox15 is expressed in these tissues, but that the enzyme may not be present in adipose tissue, testis, and heart. In contrast, activity assays with homogenate supernatants prepared from these tissues of aP2-ALOX15 mice indicated dominant 15-HETE formation by the homogenate supernatants of adipose tissue, spleen, and lungs, whereas only minor 15-HETE formation was observed in testis and heart. Taken together, our ex vivo activity data indicated that the transgenic human ALOX15 is expressed at high levels in adipose tissue but also in spleen and lungs.

### 2.6. In Vivo Activity of Transgenic Human ALOX15

Our ex vivo activity assays indicated the expression of the catalytically active transgenic human ALOX15 in different tissues, but the data did not prove the in vivo activity of the enzyme. If the transgenic enzyme is catalytically active in vivo and if this in vivo activity is mirrored on blood plasma levels of 15-HETE and other omega-6 oxygenation products of polyenoic fatty acids, the plasma concentrations of 15-HETE, 15-HEPE, 17-HDHA, and 15-HETrE should be higher in aP2-ALOX15 mice when compared with wildtype controls. To test this hypothesis, we analyzed the plasma oxylipidomes of aP2-ALOX15 mice and corresponding wildtype controls and quantified the plasma concentrations of oxygenated polyenoic fatty acids, including the major ALOX15 products [83].

As negative controls, we also quantified the plasma concentrations of other oxylipins that are not formed from arachidonic acid, eicosapentaenoic acid, docosahexaenoic acid, nor 8,11,14-eicosatrienoic acid by human ALOX15 (8-HETE, 8-HEPE, 10-HDHA, 8-HETrE). First, we quantified the sum of all oxygenated polyenoic fatty acids present in the blood plasma of the two genotypes (Figure 5A). Here, we did not find a significant difference between the two genotypes. These data indicate that the degree of oxidative challenge is similar in both genotypes. In other words, overexpression of human ALOX15 did not lead to an increased oxidative stress in the aP2-ALOX15 mice. When we analyzed the major arachidonic acid oxygenation products, we found that the plasma concentrations of 15-HETE in transgenic aP2-ALOX15 mice were almost five-fold higher than those in wildtype control animals (C57BL/6). In contrast, there were no significant differences between the two genotypes when the plasma levels of 8-HETE (not an ALOX15 product) were compared. These data can be interpreted as an indication of the in vivo activity of the transgenic human ALOX15.

Next, we analyzed the oxygenation products of three other polyenoic fatty acids. Human ALOX15 converts 5,8,11,14-eicosapentaenoic acid predominantly to 15-HEPE [83]. We found that the 15-HEPE plasma concentrations were more than five-fold higher in aP2-ALOX15 mice when compared with wildtype controls. Here again, we did not find any difference between the two genotypes for 8-HEPE, which is not formed by human ALOX15 (Figure 6C). Similar results were obtained for the major oxygenation products formed from 4,7,10,13,16,19-docosahexaenoic acid (Figure 6D) and 8,11,14-eicosatrienoic acid (Figure 6E). Here, the differences between aP2-ALOX15 mice and C57Bl/6 control animals were even more pronounced.

In summary, our plasma oxilipidomes suggest the in vivo activity of the transgenic human ALOX15. Interestingly, the in vivo catalytic activity of the transgenic ALOX15 did not induce an elevated oxidative challenge in the transgenic animals. Obviously, the reductive capacity of the ALOX15-expressing cells is high enough to ensure the instantaneous reduction of the hydroperoxy fatty acids formed by the transgenic enzyme.

### 2.7. Reproduction Statistics

ALOX15 has been implicated in spermatogenesis [84], and ALOX15^−/−^ mice are sub-fertile [85,86]. Although we did not find dramatic overexpression of the catalytically active transgene in testis (Figure 7), we compared the reproduction statistics of the aP2-ALOX15 mice with those of wildtype controls. Here, we found that the frequency of pregnancy (litters per female and month) and the reproduction efficiency (litters per female and months) were significantly elevated in aP2-ALOX15 transgenic mice (Figure 7). For the other readout parameters, no significant differences were observed when the two genotypes were compared (Figure 7). In summary, one can conclude that the aP2-ALOX15 mice, which express human ALOX15 in addition to the endogenous mouse ALOX15, were slightly more fertile than wildtype controls, although the differences were rather subtle.

### 2.8. Body-Weight Kinetics

To explore whether systemic overexpression of human ALOX15 impacts the development of mice during adolescence and adulthood, we next profiled the body-weight kinetics of aP2-ALOX15 mice and wildtype control animals starting at the age of 8 weeks. For female individuals, we found that at 8 weeks, aP2-ALOX15 mice were significantly leaner than wildtype controls (Figure 8A). In fact, between 8–30 weeks, the curve of the body-weight kinetics of aP2-ALOX15 mice was consistently below the curve of the wildtype controls, and this difference was statistically highly significant (Wilcoxon test, *p* < 0.0085). In contrast, between 31–38 weeks, no significant difference was observed between the two genotypes. These data suggest that female aP2-ALOX15 mice gained less body weight than wildtype controls during the early developmental period but that the transgenic individuals caught up with the wildtype controls at later developmental stages (Figure 8A).

When the body weights of male individuals were profiled, an inverse situation was observed. Here, we did not find significant differences between the two genotypes in early developmental stages (Figure 8B). In fact, between 8 and 20 weeks, the curves of the body-weight kinetics were superimposable, and using the Wilcoxon test, we did not observe a significant difference between the two genotypes. However, at later developmental stages (20–38 weeks), aP2-ALOX15 mice gained significantly more body weight that wildtype controls (Figure 8B). Although the extent of this difference was rather subtle, it was statistically highly significant (*p* = 0.0005). When the Wilcoxon test was applied for the entire experimental timescale, highly significant differences between aP2-ALOX15 mice and wildtype controls were observed for either sex (*p* < 0.0001), but the net effects were opposite in males and females. In females, aP2-ALOX15 mice gained less body weight, whereas male aP2-ALOX15 individuals gained more. The mechanistic basis for the observed gender specific effects have not been explored. We speculate that the transgenic expression of the ALOX15 in adipose tissue might impact the production of sexual hormones and/or of leptin in the adipose tissue. To test this hypothesis, additional experiments must be carried out that exceed the frame of the present study.

## 3. Discussion

### 3.1. Degree of Novelty and Limitations

Mammalian ALOX15 orthologs have been implicated in the differentiation of adipocytes [87], in the oxidative metabolism of fat cells [88], and in the remodeling of the adipose tissue [77]. ALOX15 mRNA expression was dramatically upregulated in white epididymal adipocytes when wildtype mice were fed a high-fat diet [87]. In 3T3 preadipocytes, ALOX15 is virtually absent, but its expression is strongly upregulated when cells were differentiated into adipocytes [87]. When treated with the ALOX15 products, these cells adopt a proinflammatory phenotype and lose their insulin resistance [9,87]. ALOX15^−/−^ mice are resistant to the induction of type-1 diabetes [89] and also to the inflammatory effects of obesity induced by a Western-type diet [90]. ALOX15-deficient nonobese diabetic mice developed diabetes at a markedly reduced rate, demonstrated improved glucose tolerance, reduced severity of insulitis, and improved beta-cell mass when compared with age-matched nondiabetic ALOX15-sufficient controls. These results suggest an important role for ALOX15 in the pathogenesis of autoimmune diabetes [91]. In most of these studies, loss-of-function strategies were employed to evaluate the role of ALOX15 in the pathogenesis of obesity and diabetes, but the application of gain-of-function strategies was rare. To address this problem, we here created transgenic mice that overexpress human ALOX15 under the control of the aP2 promoter. Our qRT-PCR studies (Figure 3) and our ex vivo activity data (Figure 5) indicate the expression of the transgene in different types of adipose tissue but also in other mesenchymal cells such as bone marrow, spleen, and peritoneal macrophages. Although our data indicate that transgene expression may not be specific for adipocytes, we did not detect transgene expression in other major organs of our aP2-ALOX15 mice, such as in liver, skin, bones, kidney, or skeleton muscles.

If one compares the aP2-ALOX15 mice created in this study with previously described ALOX transgenic mouse lines, the advantages and disadvantages of the aP2-ALOX15 mice can be summarized as follows: 

(i) Expression of the transgene is limited to a small number of special cell types, and thus, these mice are particularly suited for further investigations into the role of the ALOX15 pathway in adipocytes (Figure 5) and in hematopoietic cells (Table 2). In other studies, expression of the ALOX transgenes was controlled by different promoters directing transgene expression to other cell types [56,59,62,63,66]. Thus, for studies on the potential role of the ALOX15 pathway in adipocytes and hematopoietic cells, the previously created ALOX15 transgenic mouse lines are less suitable. On the other hand, the aP2-ALOX15 mice may not be useful to study the metabolic role of this enzyme in endothelial cells, epithelial cells, and/or macrophages. For such experiments, transgenic ALOX15 mice should be used, in which transgene expression is controlled by the preproendothelin [56], the lysozyme [63], scavenger receptor A [62], or the villin [66] promoter. 

(ii) In all previously created ALOX15 transgenic mouse lines, incorporation of the transgene into the genome was not controlled. Thus, multiple copies of the transgene might have been inserted, and incorporation of the transgene might have disrupted other genes. For the aP2-ALOX15 mice, we characterized the site of transgene insertion and found that the ALOX15 transgene was incorporated as a single-copy gene into the E1-2 region of chromosome 2 (Figure 2). Moreover, complete genome sequencing suggested that transgene incorporation did not structurally disturb other genes. 

(iii) In most previously created ALOX15 transgenic mouse lines, the catalytic activity of the transgenic enzyme was not tested, and thus, it was unclear whether the transgenic enzyme was catalytically active. For the aP2-ALOX15 mice, we carried out ex vivo ALOX15 activity assays with different cells and tissues and showed catalytic activity of the transgene (Figure 4 and Figure 5, Table 2). Moreover, we found that the product pattern formed from exogenously added arachidonic acid was very similar to that formed by recombinant human ALOX15 [79]. 

(iv) Although our ex vivo activity assays indicated the principle catalytic activity of the transgenic enzyme, such assays do not prove the in vivo activity. To show the in vivo activity of the transgenic ALOX15, we analyzed the plasma oxylipidomes (Figure 6) and found that in the blood plasma of the aP2-ALOX15 mice, the classical ALOX15 products formed from different polyenoic fatty acid were elevated. In contrast, there was no difference between aP2-ALOX15 mice and wildtype controls when unrelated oxylipins (e.g., 8-HETE, 8-HEPE, 8-HeTrE) were compared. These data suggest the in vivo activity of the transgenic enzyme. Corresponding experiments have not been carried out with any of the other ALOX transgenic mouse lines. 

(v) ALOX15 has been implicated in spermatogenesis, and ALOX15^−/−^ mice are sub-fertile [84]. Thus, there is the possibility that transgenic overexpression of ALOX15 might impact the reproduction behavior of aP2-ALOX15 mice. To test the fertility of these animals, we evaluated the reproduction statistics and found no dramatic difference with wildtype mice (Figure 7). Similar experiments have not been carried out for any of the other transgenic ALOX15 mice. 

(vi) aP2-ALOX15 mice showed gender-specific differences to wildtype controls when their body-weight kinetics were evaluated (Figure 8). This observation is not trivial and must be considered in the interpretation of future experimental data obtained with these mice in animal disease models. Here again, body-weight kinetics have not been reported for any of the other transgenic mouse lines. 

In summary, one can conclude that the aP2-ALOX15 mice created in this study constitute the most comprehensively characterized transgenic ALOX15 mouse line currently available.

The aP2-ALOX15 mice may also be used for rescue experiments reversing the effects induced by systemic or tissue-specific ALOX15 knockout. We recently reported that systemic inactivation of the ALOX15 gene induced subtle defects in the erythropoietic systems in ALOX15^−/−^ mice, as indicated by significantly reduced Hb, HK, and Ery counts [92]. When the ALOX15^−/−^ mice were crossed with our aP2-ALOX15 mice, we found that this defective phenotype was rescued, since the above-mentioned erythropoietic parameters were normalized [92]. From these data, we concluded that overexpression of human ALOX15 in hematopoietic cells may compensate for ALOX15 deficiency.

Originally, we created these mice in order to study the role of ALOX15 in adipocytes. In previous cellular studies, different ALOX-isoforms and their metabolites have been implicated in adipogenesis and in the pathogenesis of the metabolic syndrome [9,93,94,95], which is associated with hyperplasia of the adipose tissue. The aP2-ALOX15 mice appear to be a valuable research tool to test these hypotheses in vivo. The present paper describes the production and basic functional characterization of these mice, which can later be used in different animal disease models associated with adipocyte hyperplasia. Since our lab is not specialized in such diseases, and since we do not have the suitable model systems, the aP2-ALOX15 mice may be employed by interested scientists in the frame of scientific collaboration.

### 3.2. Human vs. Mouse ALOX15 as Transgene

When we started this project, we had a long discussion regarding whether we should use the endogenous mouse Alox15 or the corresponding human ortholog (ALOX15) as the transgene. This discussion was prompted by the catalytic differences of the two ALOX15 orthologs. Human ALOX15 converts arachidonic acid mainly to 15*S*-HETE (90%), and only about 10% is formed as 12*S*-HETE [79,96]. Under identical experimental conditions, mouse Alox15 exhibits an inverse product pattern. Here, 12S-HETE is dominant, whereas 15S-HETE is a minor side product [82]. The molecular basis for this difference in the reaction specificity has been explored [97,98,99,100], and the Triad Concept [26,101] has been developed as a mechanistic tool for predicting the reaction specificity of mammalian ALOX15 orthologs on the basis of their primary structures. Moreover, the evolutionary hypothesis of mammalian ALOX15 specificity [102,103] suggests that ALOX15 orthologs of those mammalian species ranked in evolution above gibbons, including humans, chimpanzees, and orangutans, express arachidonic acid 15-lipoxygenating ALOX15 orthologs. In mammals ranking in evolution below gibbons, arachidonic acid 12-lipoxygenating ALOX15 orthologs are present. Thus, the vast majority (<95%) of mammals express an arachidonic acid 12-lipoxygenating ALOX15 ortholog, despite their annotation as ALOX15. However, several mammals (about 5%) including rabbits [104], mountain hares [103], kangaroo rats [105], anteaters, and bamboo rats [103] violate this concept. Thus, because of the dominance of arachidonic acid 12-lipoxygenating ALOX15 orthologs in mammals, the endogenous mouse ALOX15 should be employed as the transgene. However, the advantage of using the human ALOX15 as transgene is that the catalytic activity of this transgene can easily be followed. In mice, there is no arachidonic acid 15-lipoxygenating ALOX-isoform and thus, 15-HETE formation can be considered as a metabolic footprint of the transgene. In contrast, several mouse ALOX-isoforms, including the endogenous ALOX15, convert arachidonic acid to 12-HETE, and thus, profiling 12-HETE formation does not allow metabolic profiling of the transgene. Thus, we decided to use human ALOX15 as the transgene.

Despite their different reaction specificity with arachidonic acid and 4,7,10,13,16,19-docosahexaenoic acid [83], mouse Alox15 and its human ortholog are very similar. They share a high degree (>85%) of amino acid sequence identity, and both enzymes are capable of oxygenating linoleic acid. For both enzymes, 13-H(p)ODE was identified as the dominant linoleic acid oxygenation product. Similarly, from 5,8,11,14,17-eicosapentaenoic acid, 15-H(p)EPE is formed as the major oxygenation product by the two ALOX15 orthologs [83]. Moreover, both ALOX15 orthologs are capable of oxygenating biomembranes, although human ALOX15 is somewhat more efficient. If ALOX15 orthologs fulfill their biological functions via the formation of specific oxygenation products from arachidonic acid or docosahexaenoic acid, there must be a functional difference between mouse and human ALOX15. In contrast, when product formation from linoleic acid and/or 5,8,11,14,17-eicosapentaenoic acid is more important, both enzymes should induce similar biological effects. If the biological functions of the ALOX15 orthologs are related to their ability to oxygenate complex substrates, there may not be major differences between mouse and human ALOX15.

For clarity, we would like to discuss the following example. Rabbit ALOX15 has been implicated in late erythropoiesis [46]. When synthesized, the enzyme oxygenates mitochondrial membrane lipids, which initiates the maturational proteolytic breakdown of the mitochondria in mature reticulocytes. If this concept is transferred to other mammals, there should not be a major impact whether the ALOX15 ortholog is an arachidonic acid 12-lipoxygenating or an arachidonic acid 15-lipoxygenating enzyme. As long as the enzyme is capable of oxygenating the membrane lipids it will fulfill its biological function. Thus, in this case, the ability of the enzyme to oxygenate complex substrates is more important for the biological function than the reaction specificity with arachidonic acid. Moreover, linoleic acid is the major polyenoic fatty acid of mitochondrial membranes. Thus, the ability of mouse and human ALOX15 to oxygenate this substrate may be more important for the biological role of the enzyme than their reaction specificity with free arachidonic acid.

### 3.3. ALOX15 Expression in Hematopoietic Cells Suppresses the Pro-Inflammatory ALOX5 Pathway

In mouse bone marrow cells, several ALOX-isoforms (ALOX15, ALOX12, ALOX5) are constitutively expressed, and thus, these cells may be good models for the exploration of functional ALOX interaction. When these cells were incubated ex vivo with arachidonic acid (Table 2), we found that 12-HETE was dominant. In addition, small amounts of 15-HETE were also detected, whereas 5-HETE formation was minimal. The most plausible explanation for this product pattern was that 12-HETE formation may be due to the catalytic activity of both endogenous ALOX12 and ALOX15. Mouse ALOX12 exclusively produces 12-HETE, whereas the endogenous ALOX15 forms 15-HETE as a minor side product. When we knocked out ALOX15 expression, the relative share of 12-HETE formation was strongly reduced, and 15-HETE formation completely disappeared. Thus, in mouse bone marrow cells, endogenous ALOX15 is responsible for the formation of 15-HETE and parts of 12-HETE. Most interestingly, however, was the observation that functional inactivation of the ALOX15 pathway strongly upregulated the catalytic activity of endogenous ALOX5 (Table 2). In fact, 5-HETE was the dominant arachidonic acid oxygenation product when bone marrow cells of ALOX15^−/−^ mice were incubated ex vivo with arachidonic acid. In peritoneal macrophages (Table 1), this effect was even more pronounced. Here, ALOX15-derived 12-HETE was dominant when ALOX15-sufficient cells were employed. In contrast, exclusive 5-HETE formation was observed with ALOX15-deficient macrophages. 

These data suggest that at least in bone marrow cells and in peritoneal macrophages, a catalytically active ALOX15 suppresses the ALOX5 pathway. Expression of the human ALOX15 transgene induced a similar repressive effect as the endogenous ALOX15 (Table 1 and Table 2). The most straightforward explanation for this observation is that endogenous and transgenic ALOX15 orthologs compete with endogenous ALOX5 for the exogenous substrate. However, there are two lines of experimental evidence arguing against this explanation: (i) the affinity of human ALOX15 [79] and human ALOX5 [106] for arachidonic acid is comparable, and thus, the suppressive effect cannot be related to competition of the two enzymes for the joint substrate; (ii) when we analyzed the free arachidonic acid, which was left over after the ex vivo incubation period, we found that about half of the substrate was not converted. These data suggest suppression of the ALOX5 pathway, even though plenty of exogenous arachidonic acid was present as ALOX5 substrate. Thus, simple substrate competition may not be the major reason for the suppression of the ALOX5 pathway by ALOX15 expression. The molecular basis for the suppressive effect of ALOX15 expression on ALOX5 has not been explored in detail, but it may be possible that primary and/or secondary products of the ALOX15 pathway directly inhibit ALOX5. This effect may be of biological relevance, since it may explain, at least in part, the anti-inflammatory role of ALOX15 in different mouse inflammation models [63,107,108], in addition to the ALOX15-dependent formation of special pro-resolving mediators [109,110]

### 3.4. Expression of ALOX15b in aP2-ALOX15 Mice

As indicated in Table 2, transgenic expression of human ALOX15 in bone marrow cells suppressed the catalytic activity of ALOX5 in our ex vivo activity assays. Unfortunately, we did not quantify the expression levels of endogenous ALOX5 or other ALOX-isoforms such as ALOX15b. In humans, ALOX15B converts AA to the same oxygenation product (15-HETE) as ALOX15, but its mouse ortholog exhibits a different product specificity [111] with free AA (8-HETE formation). Since we did not see major amounts of 8-HETE formation in our ex vivo activity assays using peritoneal lavage and bone marrow cells (Table 2), it may be concluded that ALOX15b expression in these cells may not be very pronounced. Moreover, analyses of the plasma oxylipidomes did not reveal significant differences between aP2-ALOX15 mice and wildtype controls; these data suggest that the endogenous ALOX15b pathway may have minimally altered by our genetic manipulation.

For complex substrates, the situation is somewhat different. When nanodiscs involving AA-containing phospholipids were used as substrate for recombinant mouse and human ALOX15B orthologs, 15S-HETE-containing phospholipids were detected as major reaction products [112]. In other words, with phospholipids as substrate, mouse and human ALOX15B orthologs exhibit similar reaction specificities. Because of these observations, we cannot completely exclude, on the basis of our experimental data, the modification of ALOX15b expression in the aP2-ALOX15 mice.

The biological role of ALOX15B has not been well-defined, neither in mice nor in humans. In a recent review [113], the different hypotheses on the putative physiological and pathophysiological functions of mammalian ALOX15B orthologs were summarized, but because of the lack of systemic ALOX15b^−/−^ mice, most of these hypotheses have not been confirmed under in vivo conditions.

## 4. Materials and Methods

### 4.1. Chemicals

The chemicals used for the different experiments were obtained from the following sources: phosphate-buffered saline without calcium and magnesium (PBS) from PAN Biotech (Aidenbach, Germany); EDTA from Merck KG (Darmstadt, Germany); arachidonic acid (AA) and authentic HPLC standards of HETE-isomers (15R/S-HETE, 12S/R-HETE, 8R/S-HETE, 5S-HETE) from Cayman Chem (distributed by Biomol GmbH, Hamburg, Germany); acetic acid from Carl Roth GmbH (Karlsruhe, Germany); sodium borohydride from Life Technologies, Inc (Eggenstein, Germany); restriction enzymes from ThermoFisher (Schwerte, Germany). Oligonucleotide synthesis was performed at BioTez Berlin Buch GmbH (Berlin, Germany). Nucleic acid sequencing was carried out at Eurofins MWG Operon (Ebersberg, Germany). HPLC-grade methanol, acetonitrile, n-hexane, 2-propanol, ethanol, and water were from Fisher Scientific (Schwerte, Germany).

### 4.2. Animals

A colony of homozygous ALOX15^−/−^ mice [35] that was provided years ago by Dr. C. Funk is kept in our animal house. These mice have been back-crossed into a C57BL/6J background several times [92] and were crossed with homozygous aP2-ALOX15 mice for ex vivo activity assays using peritoneal lavage cells (Figure 4). aP2-ALOX15 transgenic mice expressing human ALOX15 under the control of the aP2 promoter were created as described in Section 2.2.

### 4.3. RNA Extraction and qRT-PCR

A total of 10–30 mg (wet weight) of different tissues were stored in RNAlater solution (Sigma-Aldrich/Merck, Taufkirchen, Germany), after which they were cut into small pieces using a scalpel and then homogenized in 400 µL of LBP buffer (Nucleospin RNA plus kit, Macherey-Nagel, Düren, Germany) using a FastPrep24 homogenizer. Cell debris was spun down, and from the homogenate supernatant, total RNA was extracted following the instructions of the vendor of the Nucleospin RNA plus kit (Macherey-Nagel, Düren, Germany). Subsequently, 500 ng of RNA was reversely transcribed using the Tetro Reverse Transcriptase kit (Meridian Bioscience, Memphis, TN, USA, distributed by BioCat GmbH, Heidelberg, Germany) and Oligo dT_18_ reagents as recommended by the vendor. qRT-PCR was performed as described before [114]. Briefly, for each target gene, specific intron-spanning amplification primer combinations were synthesized (BioTez GmbH, Berlin, Germany), and external amplification standards were prepared. The following primer combinations were used: mouse ALOX15, 5′-GTACGCGGGCTCCAACAACGA-3′ and 3′-TCTCCGGGGCCCTTCACAGAA-5′; human ALOX15, 5′-ACTGAAATCGGGCTGCAAGGGG-3′ and 3′-TGGCCCACAGCCACCATAACGG-5′. Expression of target genes was quantified using standard curves (known copy numbers of the external amplification standards) and was normalized to GAPDH expression. qRT-PCR was performed on a Rotor Gene 3000 device (Corbett Research, Mortlake, Australia). Amplification products were generated, and the progress of the amplification process was followed using the SensiMix^TM^ SYBR PCR Kit (Meridian Bioscience, Memphis, TN, USA, distributed by BioCat GmbH, Heidelberg, Germany).

### 4.4. Fluorescence In Situ Hybridization (FISH)

Prometaphase chromosomes were prepared from three 8-week-old male aP2-15LOX1 mice. Spleen tissue was disrupted in 3 mL of RPMI 1640 medium using a dounce homogenizer. Six cell-culture flasks (75 cm^2^) containing 20 mL of RPMI 1640 medium, supplemented with 10% FCS, 7.5 µg/mL concanavalin A, and 5 µg/mL LPS (both from Sigma-Aldrich/Merck, Taufkirchen, Germany) were prepared. Then, 500 µL of homogenate was added to each flask and cultured for 48 h at 37 °C under 5% CO_2_-containing atmosphere. The cultured cells were harvested, and the cell suspension was filled into four 50 mL blue-cap tubes. Cells were pelleted by centrifugation for 10 min at 1000 rpm. Each cell pellet was resuspended in 10 mL of RPMI 1640 containing 10% FCS, and the two suspensions were combined. Then, 120 µL of Colcemid was added (Karyomax stock 10 µg/mL, ThermoFisher Scientific, Schwerte, Germany), and the cell suspensions were transferred to 15 cm Petri dishes. The dishes were incubated for 10 min at 37 °C, and then the cell suspensions were transferred into 50 mL blue-cap tubes. After centrifugation for 10 min at 1000 rpm, the supernatant was discarded, 10 mL of 75 mM KCl (37 °C) was added, and the samples were incubated for 15 min at 37 °C. Afterwards, 10 droplets of ice-cold fixative (20 mL of acetic acid + 60 mL of methanol) was added to each tube, and cells were pelleted by centrifugation (10 min at 1200 rpm, 4 °C). The supernatant was discarded and all cell pellets were combined and washed three times with 20 mL of fresh fixative at 4 °C. Finally, the cells, which were effectively reduced by the previous preparation to nuclei, were resuspended in 1 mL of fixative and kept at −20 °C until further use.

For FISH, ~0.1–0.2 mL of fixative (with cells/nuclei) were applied to clean and humid slides and air-dried. During this step, spread metaphases were formed by sequential evaporation of methanol and acetic acid. Before vanishing from the slide surface, acetic acid attracts atmospheric water, and the nucleic material spreads, leading to enlarged interphase nuclei and well-spread metaphase chromosomes [115]. Slides were processed using a standard FISH procedure as previously reported [116]. For mapping of the ALOX15, cDNA was used as probe. ALOX15 cDNA was labelled by degenerate oligonucleotide primed polymerase chain reaction (DOP-PCR), incorporating biotin-dUTPs during the reaction. The ALOX15 cDNA probe was applied in a one-color FISH experiment, and the probe was either detected by avidin-tagged Spectrum Orange or Spectrum Green. Then, 20 metaphases were acquired on a Zeiss Axioplan microscope (Carl Zeiss, Jena, Germany) equipped with corresponding filters and ISIS software (MetaSystems, Altlussheim, Germany). The positions of the acquired metaphases were registered; thus, the same 20 metaphases could be evaluated again after a second FISH was conducted on the same slide using a commercial multicolor FISH probe (M-FISH) set staining all 21 different murine chromosomes in specific color combinations (“SkyPaintTM DNA Kit M-10 for Mouse Chromosomes”, Applied Spectral Imaging, Edingen-Neckarhausen, Germany). Accordingly, the chromosome in which the human ALOX15 cDNA was inserted could be identified.

### 4.5. Whole-Genome Sequencing

Genomic DNA was prepared from 58 mg of liver tissue using the Invisorb^®^ Spin Tissue Mini Kit (Invitek Molecular GmbH, Berlin, Germany). An additional RNAse treatment was performed, and the RNA-free DNA preparation was quality-checked with agarose gel electrophoresis. The DNA was sequenced using the shot-gun technology (ATLAS Biolabs GmbH, Berlin, Germany). The whole genome sequence data can be obtained by interested scientists upon request from Dr. K.R. Kakularam

### 4.6. Preparation of Bone Marrow Cells and Peritoneal Macrophages

For preparation of peritoneal macrophages, 10 mL of PBS was injected into the peritoneal cavity of sacrificed mice. The belly was gently massaged for 2 min and the fluid was removed by puncturing the peritoneal cavity. Usually, about 8–9 mL of cell suspension was recovered. Cells were spun down for 15 min at 800 g and were washed twice with PBS. Finally, the cells were reconstituted in 0.5 mL of PBS and were used for ex vivo ALOX activity assays. To prepare bone marrow cells, mice were sacrificed under anesthesia by cervical dislocation, and the two femur bones were prepared. The ends of the bones were cut off, and the bone marrow cavity was rinsed with 10 mL PBS. The cell suspensions were combined, and cells were pelleted (15 min, 800 g), washed twice with PBS, and reconstituted in 1 mL of PBS. Aliquots of this cell suspension were used for ex vivo ALOX activity assays quantifying the formation of oxygenated AA derivatives.

### 4.7. Ex Vivo Activity Assays

To explore whether the transgenic human ALOX15 is expressed in different cells and tissues as a catalytically active enzyme, we carried out ex vivo activity assays using tissue homogenate or cell suspensions (peritoneal macrophages, bone marrow cells) as enzyme source. For the activity assays of solid tissues, 200 mg (wet weight) of tissue were homogenized in 2 mL of PBS using the Fast Prep-24 homogenizer (MP Biomedicals, Eschwege, Germany). The tissue homogenates were centrifuged for 10 min at 15,000× *g* (4 °C), and the homogenate supernatants were used as enzyme source. Aliquots (20–200 µL depending on the protein content) of the homogenate supernatants were incubated at room temperature in 1 mL of PBS containing 100 µM of arachidonic acid for 15 min. The reaction was terminated by the addition of 1 mg of solid sodium borohydride. After the addition of 35 µL of acetic acid, the lipids were extracted twice with 1 mL of ethyl acetate. Then, 1 mL of 2-propanol was added, and the solvents were evaporated in a rotatory evaporator. The remaining lipids were reconstituted in 0.5 mL of RP-HPLC column solvent (acetonitrile:water:acetic acid, 70:30:0.1, by vol.), the sample was sonicated, debris was spun down, and aliquots were injected for RP-HPLC quantification of the ALOX15 products. A similar method was employed for the ex vivo activity assays of the cell suspensions. For these assays, 1–10 × 10^6^ cells were incubated at room temperature in 0.5 mL of PBS containing 100 µM of arachidonic acid. After 10 min, the hydroperoxy fatty acids that formed were reduced by the addition of 1 mg of solid sodium borohydride, the sample was acidified, and 0.5 mL of ice-cold acetonitrile was added. The protein precipitate was spun down, and aliquots of the protein-free supernatant were injected to RP-HPLC for quantification of the hydroxy fatty acids.

### 4.8. RP-HPLC Analysis of the ALOX Products

To quantify the amounts of ALOX products formed during the incubation period of the ex vivo activity assays, a Shimadzu instrument (LC20 AD) equipped with a diode array detector (SPD M20A) was used, and the hydroxy fatty acids were separated on a Nucleodur C_18_ Gravity column (Macherey-Nagel, Düren, Germany; 250 × 4 mm, 5 μm particle size), which was coupled with a guard column (8 × 4 mm, 5 μm particle size). The analytes were eluted isocratically using a solvent system consisting of acetonitrile:water:acetic acid (70:30:0.1, by vol) with a flow rate of 1 mL/min at 25 °C. The absorbance at 235 nm (absorbance maximum of the conjugated dienes) was recorded, and the UV spectra of the dominant peaks that were recorded during the chromatographic runs were evaluated.

### 4.9. LC-MS/MS Analysis of the Blood Plasma Oxylipidomes

Mouse Alox15 is an arachidonic acid 12-lipoxygenating enzyme [82], whereas the human ortholog oxygenates the same substrate predominantly to 15-H(p)ETE [78]. If aP2-ALOX15 transgenic mice have significantly elevated 15-HETE plasma levels, these data may be interpreted as an indication of the in vivo activity of the transgenic enzyme. To explore whether the pattern of the plasma oxylipins was impacted by in vivo expression of the transgenic human ALOX15, we quantified the amounts of more than 40 different free-oxygenated PUFAs in the blood plasma [117]. For this purpose, EDTA blood was drawn from sacrificed mice, and after a 15 min incubation period, the blood plasma was prepared by centrifugation. Then, 10 µL of blood plasma was mixed with 450 µL of water and 10 µL of a mixture of internal standards (LTB4-d4, 20-HETE-d6, 15-HETE-d8, 13-HODE-d4, 14,15-DHET-d11, 9,10-DiHOME-d4, 12,13-EpOME-d4, 8,9-EET-d11, PGE2-d4; 10 ng/mL each). Next, 5 µL of a butylhydroxytoluene (BHT) solution were added to prevent PUFA autooxidation during sample workup and storage. Plasma proteins were precipitated by the addition of 100 µL of a 1:4 mixture (by vol.) of glycerol/water and 500 µL of acetonitrile. The pH was adjusted to 6.0 by the addition of 2 mL of phosphate buffer (0.15 M), the precipitated proteins were removed by centrifugation, and the clear supernatant was used for solid-phase lipid extraction on a 200 mg Agilent Bond Elut Certify II cartridge (Agilent Technologies, Santa Clara, USA). Before sample application, the cartridge was conditioned with 3 mL of methanol and 3 mL of phosphate buffer (0.15 M, pH 6.0). After the sample was applied, the column was washed with 3 mL of a 1:1 mixture (by vol.) of methanol: water, and the oxygenated fatty acids were eluted with a 74:25:1 mixture (by vol.) of ethyl acetate: n-hexane:acetic acid. The solvents were evaporated in a stream of nitrogen, and the remaining lipids were reconstituted in 100 µL of a 6:4 mixture (by vol.) of methanol: water and used for LC-MS/MS analysis. The chemical identity of the different analytes was concluded from co-chromatography with authentic standards, and for each of the quantified metabolites, a calibration curve was established.

LC-MS/MS was carried out on an Agilent 1290/II LC-MS system consisting of a binary pump system, an autosampler, and a column oven (Agilent Technologies, Waldbronn, Germany). As a stationary phase, we employed an Agilent Zorbax Eclipse C_18_ UPLC column (150 × 2.1 mm, 1.8 µm particle size). The column temperature was set at 30 °C. As a mobile phase, we used a solvent gradient that was mixed from two stock solutions. Stock A: water containing 0.05% acetic acid. Stock B: 1:1 mixture (by vol.) of methanol: acetonitrile. The HPLC system was connected to a triple-quadrupole MS system (Agilent 6495 System, Agilent Technologies, Santa Clara, CA, USA). Negative electrospray ionization was carried out. The mass spectrometer was run in dynamic MRM mode, and each metabolite was detected simultaneously by two independent mass transitions that are characteristic for the different analytes. Experimental raw data were evaluated with the Agilent Mass-Hunter software package, version B10.0. For all metabolites analyzed in this study, individual calibration curves were established, and the lower detection limits were also determined. More detailed information on this analytical procedure is given in [117].

### 4.10. Reproduction Statistics

Nineteen breeding pairs (1 male, 2 females/cage) were investigated in a time frame of 4–6 months, and the following reproduction parameters were determined: litters/month, pups/litter, male/female ratio, dead pups before weaning.

### 4.11. Body-Weight Kinetics

The body weights of 5 males and 5 females (aP2-ALOX15 and C57BL/6 as controls) were determined once a week between 8 and 34 weeks of age.

### 4.12. Statistical Evaluation of the Experimental Raw Data

Statistical calculations and figure design were performed using GraphPad prism version 8.00 for Windows (GraphPad Software, La Jolla, CA, USA, (license obtained on 8 January 2021).

## 5. Conclusions

Transgenic mice expressing human ALOX15 under the control of the aP2 (activating protein 2) promoter in addition to the endogenous ALOX15 are viable and reproduce normally, but exhibited gender-specific differences with wildtype controls when their body-weight kinetics were evaluated. These mice can now be used in whole-animal disease models associated with adipose tissue hyperplasia, such as adipositas, diabetes, and metabolic syndrome.

## Figures and Tables

**Figure 1 ijms-24-04815-f001:**
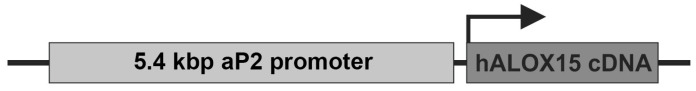
Schematic structure of the ALOX15 transgene used for the creation of aP2-ALOX15 transgenic mice. The transgene construct was microinjected into fertilized eggs of C57BL/6 mice, and heterozygous transgenic allele carriers were used to establish a colony of homozygous aP2-ALOX15 mice.

**Figure 2 ijms-24-04815-f002:**
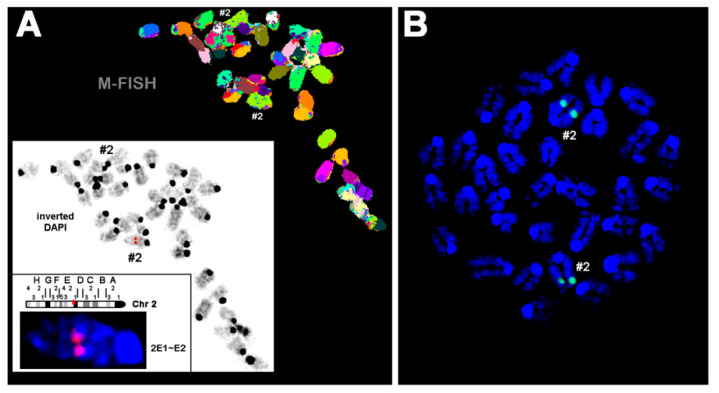
Fluorescence in situ hybridization of mouse chromosomes using the human ALOX15 cDNA as probe. Pro-metaphase chromosomes were prepared as described in Section 4. (**A**) A Metaphase spread prepared from hepatocytes of a heterozygous founder animal is shown after the M-FISH set using all 21 murine whole chromosomes. In the white frame, the inverted 4′,6-diamidino-2-phenylindole counter-staining of the same metaphase is shown with the signal of the ALOX15 cDNA (red signal) from the first round of FISH (for details see Section 4). Accordingly, the human ALOX15 cDNA was inserted in one of the two murine chromosomes 2 (#2), more exactly into subband 2E1~E2. (**B**) FISH result of a homozygous aP2-ALOX15 individual. These data indicate that the human ALOX15 transgene was inserted in both homologous murine chromosomes 2. Here, chromosomes were counterstained in blue by DAPI.

**Figure 3 ijms-24-04815-f003:**
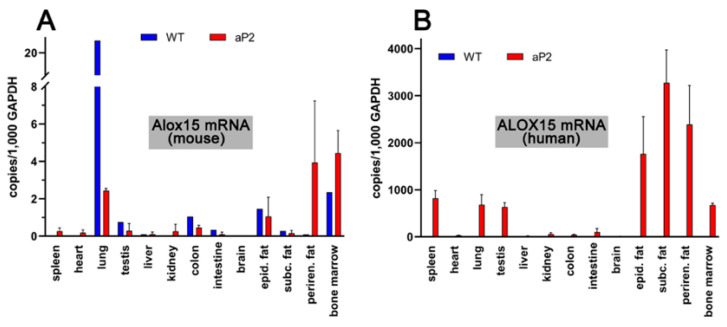
Expression of ALOX15 orthologs in different tissues of transgenic aP2-ALOX15 mice and wildtype controls (C57BL/6). (**A**) Amplification of endogenous mouse Alox15 mRNA. (**B**) Amplification of transgenic human ALOX15 mRNA. Ortholog specific primer combinations were used for these analyses; the sequences of these DNA primers are described in detail in Section 4.

**Figure 5 ijms-24-04815-f005:**
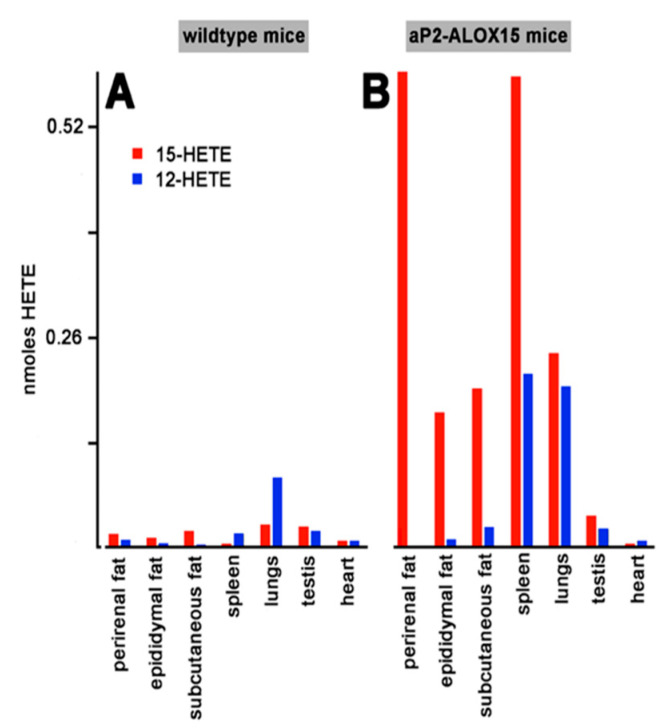
Ex vivo activity assays of solid tissues prepared from aP2-ALOX15 mice and corresponding wildtype controls. Homogenate supernatants were prepared from solid tissues of aP2-ALOX15 mice and corresponding wildtype controls. Aliquots of these supernatants (normalized to the protein content of the homogenate supernatants) were incubated in PBS with arachidonic acid as described in Section 4, and the oxygenation products were analyzed by RP-HPLC. 12-HETE and 15-HETE were separately quantified, and the area units of the two peaks were converted to nmoles HETE formation. (**A**) Wildtype mice, (**B**) aP2-ALOX15 mice.

**Figure 6 ijms-24-04815-f006:**
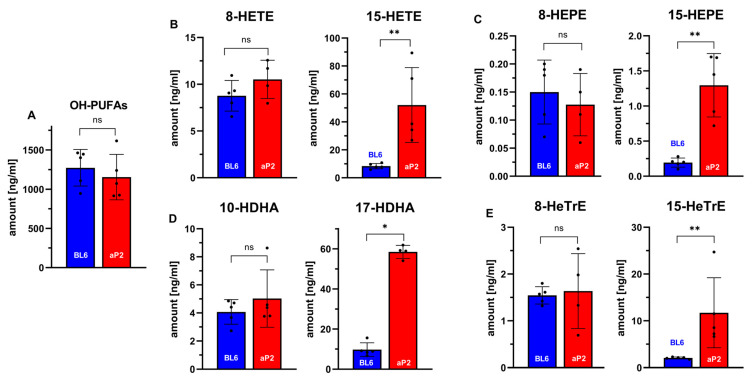
In vivo ALOX15 activity assays quantifying selected oxylipins in the blood plasma of aP2-ALOX15 mice and of wildtype controls (C57BL/6). EDTA blood was prepared from sacrificed mice (n = 5 of each genotype) by heart puncture and was incubated in vitro for 15 min at 37 °C. Blood cells were pelleted, and the plasma was shock-frozen. After thawing, total lipids were extracted and hydrolyzed, and the oxylipidomes were quantified by LC-MS/MS (see Section 4). Selected metabolites are shown. (**A**) Sum of all oxygenated fatty acids. (**B**) Selected arachidonic acid oxygenation products. (**C**) Selected 5,8,11,14,17-eicosapentaenoic acid oxygenation products. (**D**) Selected 4,7,10,13,16,19-docosahexaenoic acid oxygenation products. (**E**) Selected 8,11,15-eicosatrienoic acid oxygenation products. ns, statistically not significant, * statistically significant with *p* < 0.05, ** statistically significant with *p* < 0.01.

**Figure 7 ijms-24-04815-f007:**
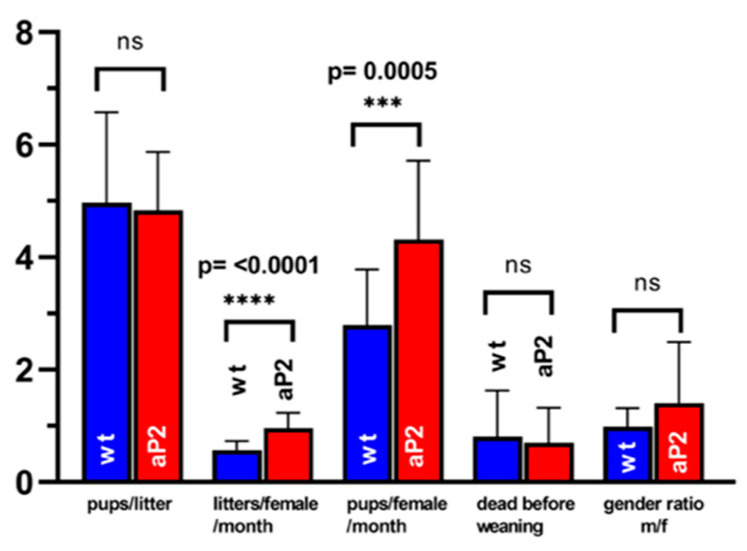
Reproduction statistics of aP2-ALOX15 mice and corresponding wildtype controls. 19 breeding pairs (two female individuals plus one male individual) of aP2-ALOX15 mice and of wildtype control animals (C57BL6) were established and kept together for 4–6 months. Litter size (pups per litter), frequency of pregnancy (litters per female and month), reproduction efficiency (pups per female and month), newborn survival rate (death before weaning), and gender ratio (males/females) were determined as readout parameters to compare the reproduction kinetics of the genetically modified aP2-ALOX15 mice with wildtype control animals. ns, statistically not significant, *** statistically significant with *p* < 0.001, **** statistically significant with *p* < 0.0001.

**Figure 8 ijms-24-04815-f008:**
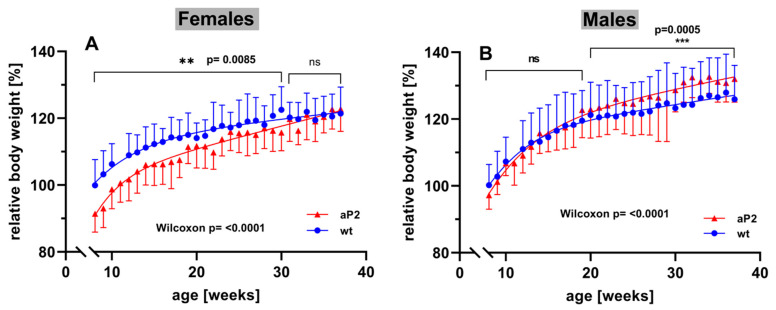
Body-weight kinetics of aP2-ALOX15 mice and corresponding wildtype controls. Female and male aP2-ALOX15 mice (n = 10 of either sex) and wildtype control animals (n = 10 of either sex) were grouped in separate cages (a single cage for each gender and sex, four cages altogether) and received water and standard chow diet *ad libitum*. Individual body weights were determined once a week. The mean of the body weights of the wildtype individuals entering the experiments (8 weeks old) was set at 100%, and the relative body weights of all individuals were calculated throughout the entire time course of the experiment. (**A**) female individuals, (**B**) male individuals. ns, statistically not significant, ** statistically significant with *p* < 0.01, *** statistically significant with *p* < 0.001.

**Table 2 ijms-24-04815-t002:** Ex vivo activity assays of bone marrow cells prepared from mice of different genotype. 5 × 10^7^ bone marrow cells prepared from mice of different genotype (n = 3 of each genotype) were incubated with exogenously added arachidonic acid as described in Section 4. The arachidonic-acid-derived oxygenation products were prepared and further analyzed by RP-HPLC (see Section 4). Statistical evaluation of the experimental raw data was performed using the Student’s t-test function of the Microsoft Excel software package. ^#^ wildtype vs. aP2-ALOX15, *p* < 0.01; ^§^ aP2-ALOX15 vs. aP2-ALOX15 + ALOX15^−/−^, *p* < 0.01.

Genotype	Relative Share of Different HETE Isomers (%)		
	15-HETE	12-HETE	5-HETE
wildtye	4.9 ± 1.6 ^#^	93.8 ± 3.6 ^#^	1.3 ±2.2
ALOX15^−/−^	0	37.7 ± 0.3	62.3 ± 0.3
aP2-ALOX15	55.5 ± 1.6 ^#,§^	44.4 ± 1.6 ^#,§^	0
aP2-ALOX15 + ALOX15^−/−^	70.3 ± 0.9 ^§^	10.9 ± 0.8 ^§^	2.7 ± 0.5

## Data Availability

All data generated or analyzed during this study are included in this published article. The mass spectral raw data of the plasma oxylipidoms can be obtained upon request from D. M. Rothe (Lipidomix GmbH, Robert-Roessle-Str. 10, 13125 Berlin, Germany; email: michael.rothe@lipidomix.de).

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
