# Peer review of "Functional Characterization of Transgenic Mice Overexpressing Human 15-Lipoxygenase-1 (ALOX15) under the Control of the aP2 Promoter"

_ijms, 2023, doi:10.3390/ijms24054815_

Round 1

Reviewer 1 Report

In mouse arteries, Alox15 [leukocyte-type 12/15-lipoxygenase (LO)] is assumed to regulate vascular function by metabolizing arachidonic acid (AA) to dilator eicosanoids that mediate the endothelium-dependent relaxations to AA and acetylcholine (ACh). However, Alox15 deficiency increased infiltration of proinflammatory macrophages and upregulated proinflammatory and necroptotic signaling in dermal adipose tissue in the dorsal skin. In this study, the authors created transgenic mice (aP2-ALOX15 mice) expressing human ALOX15 under the control of the aP2 (adipocyte fatty acid binding protein 2) promoter. This is an interesting study and is well executed. For the benefit of the reader, however, the authors should be more clarification/mention as following points.

Major concern:

1.     The advantage and disadvantage of aP2-ALOX15 mice should be described more detail when aP2-ALOX15 mice were used as experimental animal model.

2.     In Abstract section: “The aP2-ALOX15 mice characterized here can now be used for gain-of-function studies evaluating the biological role of ALOX15”. How to prove it? Did the authors compare the wildtype on biological role? It is still not enough to describe the biological function if only according to the results of this study.

3.     There is no Conclusion section.

4.     Please describe the biological functions of ALOX15B in aP2-ALOX15 mice.

Author Response

Dear editor,

on behalf of all co-authors, I should like to thank the two reviewers for critical reading of our ms. Their comments were valuable and constructive and revision of the ms according to their points of criticisms led to improvement. The critical remarks of the reviewers are addressed in this rebuttal letter on the point-to-point basis and the modifications introduced into the ms during revision are clearly labeled (yellow background) in the labeled version of the revised ms. For the revised version of the ms we modified the text and updated the reference list. In addition to the labeled version of the revised ms, which will be submitted in form of a supplemental file, we will submit an unlabeled version of the revised ms that can be used for publication.

Reviewer 1

Remark 1 of reviewer 1: The advantage and disadvantage of aP2-ALOX15 mice should be described more detail when aP2-ALOX15 mice were used as experimental animal model.

Response of authors: The aP2-ALOX15 mice express the AA 15-lipoxygenting human ALOX15 in hematopoietic cells and adipocytes. In cells of other origin (epithelial cells, endothelial cells) transgene expression is minimal. Thus, these mice are particularly useful for investigations on the role of the ALOX15 pathway in adipocytes and hematopoietic cells. They may not be of particular interest for studies on the metabolic role of this pathway in epithelial or endothelial cells. This statement was included into the ms (page 14, lines 510-514).

Remark 2 of reviewer 1: In Abstract section: “The aP2-ALOX15 mice characterized here can now be used for gain-of-function studies evaluating the biological role of ALOX15”. How to prove it? Did the authors compare the wildtype on biological role? It is still not enough to describe the biological function if only according to the results of this study.

Response of authors: Originally, we have created these mice in order to study the role of ALOX15 in adipocytes. In previous cellular studies different ALOX-isoforms and their metabolites have been implicated in adipogenesis and in the pathogenesis of the metabolic syndrome, which is associated with hyperproliferation of the adipose tissue. To test these hypotheses in vivo the aP2-ALOX15 mice appear to be a valuable research tool. The present paper describes the production and basic functional characterization of these mice, which will later on be used in different animal disease models related with adipocyte hyperproliferation. This paragraph was introduced into the ms on p. 14, line 523-528.

Remark 3 of reviewer 1: There is no Conclusion section.

Response of authors: As requested by the reviewer we included a conclusion section into the revised ms. It reads (p. 21, line 820-824): Transgenic mice expressing the arachidonic acid 15-lipoxygenating human ALOX15 under the control of the aP2 (activating protein 2) promoter in addition to their arachidonic acid 12-lipoxygenating endogenous enzyme are viable, reproduce normally but they exhibit gender-specific differences to wildtype controls when their body weight kinetics were evaluated during adolescence and early adulthood.

Remark 4 of reviewer 1: Please describe the biological functions of ALOX15B in aP2-ALOX15 mice.

Response of authors: We thank the reviewer for making this point. In general, the biological role of ALOX15B has not clearly been identified yet. In an excellent recent review, the different hypotheses on the possible physiological and patho-physiological functions of the ALOX15B have been summarized but because of the lack of alox15b knockout mice most of these data have not been confirmed or disconfirmed under in vivo conditions. We describe this situation in a separate paragraph that has been introduced during revision into the ms (page 16-17, line 627-645).

Reviewer 2 Report

In this manuscript, transgenic mice with over-expression of human ALOX15 under the control of the aP2 promoter were created. The obtained transgenic mice were characterized.

Transgenic mice with ALOX15 knockout or over-expression have been created by many researchers. Using variable lines of transgenic mice, the phenotypes and disease-related ALOX15 have been characterized. The authors had created another line of transgenic mice with ALOX15 over-expression. However, there was no disease-related phenotype characterized. Currently, the scientific value of this new-added line of ALOX15 transgenic mice is limited.

1.     Since there are many transgenic mice existing, the advantages and further utilities of this new line should be highlighted.

2.     Other than genetic identification, biological identification and implication should be performed to express scientific value of this line of mice.

3.     ALOX15 has been implicated in the metabolism of lipid and shows effects on adipose tissues. Thus, lipid metabolism and adipogenic changes should be determined in this line of mice.

Author Response

Dear editor,

on behalf of all co-authors, I should like to thank the two reviewers for critical reading of our ms. Their comments were valuable and constructive and revision of the ms according to their points of criticisms led to improvement. The critical remarks of the reviewers are addressed in this rebuttal letter on the point-to-point basis and the modifications introduced into the ms during revision are clearly labeled (yellow background) in the labeled version of the revised ms. For the revised version of the ms we modified the text and updated the reference list. In addition to the labeled version of the revised ms, which will be submitted in form of a supplemental file, we will submit an unlabeled version of the revised ms that can be used for publication.

Reviewer 2

Remark 1 of reviewer 2: Since there are many transgenic mice existing, the advantages and further utilities of this new line should be highlighted.

Response of authors: We follow the advice of the reviewer and highlight the advantages of this new transgenic mouse strain and describe the possible application of these mice in animal disease models in more detail (p. 14, lines 510-514).

Remark 2 of reviewer 2: Other than genetic identification, biological identification and implication should be performed to express scientific value of this line of mice.

Response of authors: We do agree with the reviewer that creation and basic functional characterization of the aP2-ALOX15 mice per se does not give these mice a high scientific value. This value will hopefully be indicated later on, when these mice are used in animal disease models associated with abnormal metabolic functions of the adipose tissue. Such experiments require specific research tools and the aP2 mice created here might be suitable for such purpose. Because of the comprehensive physiological characterization of these mice interpretation of the outcome of these studies are more straightforward. This is stressed in the last sentence of the Summary.

Remark 3 of reviewer 2: ALOX15 has been implicated in the metabolism of lipid and shows effects on adipose tissues. Thus, lipid metabolism and adipogenic changes should be determined in this line of mice.

Response of authors: This conclusion is straightforward and we completely agree with the reviewer. The aP2-ALOX mice should be valuable tools to explore the patho-mechanisms of diseases associated with aberrant adipose tissue metabolism. Unfortunately, we are no experts in this field of research but we hope that publication of our data might prompt interested adipocyte experts to contact us so that collaborative experiments can be performed (see page 14, line 510-514).

We hope that we have adequately addressed the concerns of the reviewers and that the ms in its revised version may be acceptable for publication.

Sincerely,

Round 2

Reviewer 1 Report

This manuscript is now suitable for publication. 

Author Response

We thank the reviewer for critical reading of the revised ms. No changes have been requested.

Reviewer 2 Report

The raised concerns had not been responded well. As the first round review, only basic characterization of new line of transgenic mouse is not helpful to the reader.  The advantages and disadvantages among new line and well-known lines should be compared. Besides, disease-related characterization and investigation is important to the demonstration of workful mouse line.

Author Response

we once again would like to thank the reviewers for critical reading of the ms. Reviewer 1 did not have any additional requests for textual modifications. In contrast, reviewer 2 reported that the authors did not well respond to his (her) points of criticisms. 1. He (she) feels that the advantages and disadvantages between the new transgenic mouse line and previously published ALOX transgenic mice should be compared. 2. He (she) criticized that no disease disease-related characterization has been carried out.

  1. With all due respect for the expertise of this reviewer, we cannot follow the first point of criticism. In the revised version of the ms we clearly describe the advantages and disadvantages of the new transgenic mouse line (labeled paragraphs on page 14). Although we feel that the statements made in the revised version of the ms are sufficient to address this point, we follow the advice of the reviewer and extended this paragraph in the rerevised version of the ms. This paragraph (p. 14-15, lines 512-558) now reads: If one compares the aP2-ALOX15 mice created in this study with previously described ALOX transgenic mouse lines the following advantages and disadvantages of the aP2-ALOX15 mice can be summarized: i) Expression of the transgene is limited to a small number of special cell types and thus, these mice are particularly suited for further investigations into the role of the ALOX15 pathway in adipocytes (Figure 5) and in hematopoietic cells (Table 2). In other studies, expression of the ALOX transgenes was controlled by different promoters directing transgene expression to other cell types (56, 59, 62, 63, 66). Thus, for studies on the potential role of ALOX15 pathway in adipocytes and hematopoietic cells the previously created ALOX15 transgenic mouse lines are less suitable. On the other hand, the aP2-ALOX15 mice may not be useful to study the metabolic role of this enzyme in endothelial cells, epithelial cells and/or macrophages. For such experiments transgenic ALOX15 mice should be used, in which transgene expression is controlled by the preproendothelin (56), the lysozyme (63), scavenger receptor A (62) or the villin (66) promoter. ii) In all previously created ALOX15 transgenic mouse lines incorporation of the transgene into the genome was not controlled. Thus, multiple copies of the transgene might have been inserted and incorporation of the transgene might have disrupted other genes. For the aP2-ALOX15 mice we characterized the site of transgene insertion and found that the ALOX15 transgene was incorporated as single copy gene into the E1-2 region of chromosome 2 (Figure 2). Moreover, complete genome sequencing suggested that transgene incorporation did not structurally disturb other genes. iii) In most previously created ALOX15 transgenic mouse lines the catalytic activity of the transgenic enzyme has not been tested and thus, it remained unclear whether the transgenic enzyme was catalytically active. For the aP2-ALOX15 mice we carried out ex vivo ALOX15 activity assays with different cells and tissues and showed catalytic activity of the transgene (Figure 4, 5, Table 2). Moreover, we found that the product pattern formed from exogenously added arachidonic acid was very similar to that formed by recombinant human ALOX15 (79). iv) Although our ex vivo activity assays indicate the principle catalytic activity of the transgenic enzyme, such assays do not prove the in vivo activity. To show the in vivo activity of the transgenic ALOX15 we analyzed the plasma oxylipidomes (Figure 6) and found that in the blood plasma of the aP2-ALOX15 mice the classical ALOX15 products formed from different polyenoic fatty acid are elevated. In contrast, there was no difference between aP2-ALOX15 mice and wildtype controls when unrelated oxylipins (e.g. 8-HETE, 8-HEPE, 8-HeTrE) were compared. These data suggest the in vivo activity of the transgenic enzyme and corresponding experiments have not been carried out with any of the other ALOX transgenic mouse lines. v) ALOX15 has been implicated in spermatogenesis and Alox15-/- mice are subfertile (85). Thus, there is the possibility that transgenic overexpression of ALOX15 might impact the reproduction behavior of the aP2-ALOX15 mice. To test the fertility of these animals we evaluated the reproduction statistics and found no dramatic differenced to wildtype mice (Figure 7). Similar experiments have not been carried out for any of the other transgenic ALOX15 mice. vi) aP2-ALOX15 mice show gender-specific differences to wildtype controls when their body weight kinetics were evaluated (Figure 8). This observation is not trivial and must be considered for the interpretation of future experimental data obtained with these mice in animal disease models. Here again, for neither of the previously transgenic mouse lines body weight kinetics have been reported. In summary, one can conclude that the aP2-ALOX15 mice created in this study constitute the most comprehensively characterized transgenic ALOX15 mouse line currently available.

2. The reviewer is correct that we did not test the aP2-ALOX15 mice in any animal disease model. This was, however, not the aim of this study. We clearly state in the ms (Title, Abstract, Discussion) that this paper is focused on creation and basic functional characterization of the aP2-ALOX15 mice. The ms involves 8 multi-panel figures and two data tables, which is according to our opinion sufficient for a separate scientific report. We do agree with the reviewer that testing of these mice in animal disease models is of interest for readers. However, corresponding experiments would clearly exceed the frame of the present study. As indicated above the aP2-ALOX15 mice are useful to study the possible role of the ALOX15 in metabolic, inflammatory, hyperproliferative and neurological diseases. In our lab we have whole animal models for different inflammatory disorders and experiments are currently underway to explore how the aP2-ALOX15 mice behave in the DSS-induced colitis model and in the CFA-induced paw edema model. Unfortunately, our lab is not specialized in hyperproliferative, metabolic and neurological diseases and we do not have the suitable model systems. However, we hope that publication the aP2-ALOX15 mouse data might prompt experts in these fields to use our mice in the frame of scientific collaborations. In summary, this paper describes the creation and comprehensive functional characterization of an interesting in vivo research tool. The application of this tool in different disease model will be explored in follow-up studies. Since we have addressed this point already in the original version of the paper no additional textual modification of the ms is required. 

Round 3

Reviewer 2 Report

There is no additional comment.